# Gated Self-Matching Networks for
# Reading Comprehension and Question Answering

## Abstract

In this paper, we present the gated self-matching networks for reading comprehension style question answering, which aims to answer questions from a given passage. We first match the question and passage with gated attention-based recurrent networks to obtain the question-aware passage representation. Then we propose a self-matching attention mechanism to refine the representation by matching the passage against itself, which effectively encodes information from the whole passage. We finally employ the pointer networks to locate the positions of answers from the passages. We conduct extensive experiments on the SQuAD dataset. The single model achieves 71.3% on the evaluation metrics of exact match on the hidden test set, while the ensemble model further boosts the results to 75.9%. At the time[1] of submission of the paper, our model holds the first place on the SQuAD leaderboard for both single and ensemble model.

## 1 Introduction

In this paper, we focus on reading comprehension style question answering which aims to answer questions given a passage or document. We specifically focus on the Stanford Question Answering Dataset (SQuAD) (Rajpurkar et al., 2016), a large-scale dataset for reading comprehension and question answering which is manually created through crowdsourcing. SQuAD constrains answers to the space of all possible spans within the reference passage, which is different from cloze-style reading comprehension datasets (Hermann et al.,

---
[1] On Feb. 6, 2017

2015; Hill et al., 2016) in which answers are single words or entities. Moreover, SQuAD requires different forms of logical reasoning to infer the answer (Rajpurkar et al., 2016).

Rapid progress has been made since the release of the SQuAD dataset. Wang and Jiang (2016b) build question-aware passage representation with match-LSTM (Wang and Jiang, 2016a), and predict answer boundaries in the passage with pointer networks (Vinyals et al., 2015). Seo et al. (2016) introduce bi-directional attention flow networks to model question-passage pairs at multiple levels of granularity. Xiong et al. (2016) propose dynamic co-attention networks which attend the question and passage simultaneously and iteratively refine answer predictions. Lee et al. (2016) and Yu et al. (2016) predict answers by ranking continuous text spans within passages.

Inspired by Wang and Jiang (2016b), we introduce a gated self-matching network, illustrated in Figure 1, an end-to-end neural network model for reading comprehension and question answering. Our model consists of four parts: 1) the recurrent network encoder to build representation for questions and passages separately, 2) the gated matching layer to match the question and passage, 3) the self-matching layer to aggregate information from the whole passage, and 4) the pointer-network based answer boundary prediction layer. The key contributions of this work are three-fold.

First, we propose a gated attention-based recurrent network, which adds an additional gate to the attention-based recurrent networks (Bahdanau et al., 2014; Rocktäschel et al., 2015; Wang and Jiang, 2016a), to account for the fact that words in the passage are of different importance to answer a particular question for reading comprehension and question answering. In Wang and Jiang (2016a), words in a passage with their corresponding attention-weighted question context are en-

coded together to produce question-aware passage representation. By introducing a gating mechanism, our gated attention-based recurrent network assigns different levels of importance to passage parts depending on their relevance to the question, masking out irrelevant passage parts and emphasizing the important ones.

Second, we introduce a self-matching mechanism, which can effectively aggregate evidence from the whole passage to infer the answer. Through a gated matching layer, the resulting question-aware passage representation effectively encodes question information for each passage word. However, recurrent networks can only memorize limited passage context in practice despite its theoretical capability. One answer candidate is often unaware of the clues in other parts of the passage. To address this problem, we propose a self-matching layer to dynamically refine passage representation with information from the whole passage. Based on question-aware passage representation, we employ gated attention-based recurrent networks on passage against passage itself, aggregating evidence relevant to the current passage word from every word in the passage. A gated attention-based recurrent network layer and self-matching layer dynamically enrich each passage representation with information aggregated from both question and passage, enabling subsequent network to better predict answers.

Lastly, the proposed method yields state-of-the-art results against strong baselines. Our single model achieves 71.3% exact match accuracy on the hidden SQuAD test set, while the ensemble model further boosts the result to 75.9%. At the time of submission of this paper, our model holds the first place on the SQuAD leader board.

## 2 Task Description

For reading comprehension style question answering, a passage **P** and question **Q** are given, our task is to predict an answer **A** to question **Q** based on information found in **P**. The SQuAD dataset further constrains answer **A** to be a continuous sub-span of passage **P**. Answer **A** often includes non-entities and can be much longer phrases. This setup challenges us to understand and reason about both the question and passage in order to infer the answer. Table 1 shows a simple example from the SQuAD dataset.

---

**Passage**: Tesla later approached Morgan to ask for more funds to build a more powerful transmitter. **When asked where all the money had gone, Tesla responded by saying that he was affected by the Panic of 1901**, which he (Morgan) had caused. Morgan was shocked by the reminder of his part in the stock market crash and by Tesla's breach of contract by asking for more funds. Tesla wrote another plea to Morgan, but it was also fruitless. Morgan still owed Tesla money on the original agreement, and Tesla had been facing foreclosure even before construction of the tower began.
**Question**: On what did Tesla blame for the loss of the initial money?
**Answer**: Panic of 1901

---

Table 1: An example from the SQuAD dataset.

## 3 Gated Self-Matching Networks

Figure 1 gives an overview of the gated self-matching networks. First, the question and passage are processed by a bi-directional recurrent network (Mikolov et al., 2010) separately. We then match the question and passage with gated attention-based recurrent networks, obtaining question-aware representation for the passage. On top of that, we apply self-matching attention to aggregate evidence from the whole passage and refine the passage representation, which is then fed into the output layer to predict the boundary of the answer span.

### 3.1 Question and Passage Encoder

Consider a question Q = $\{w_t^Q\}_{t=1}^m$ and a passage P = $\{w_t^P\}_{t=1}^n$. We first convert the words to their respective word-level embeddings ($\{e_t^Q\}_{t=1}^m$ and $\{e_t^P\}_{t=1}^n$) and character-level embeddings ($\{c_t^Q\}_{t=1}^m$ and $\{c_t^P\}_{t=1}^n$). The character-level embeddings are generated by taking the final hidden states of a bi-directional recurrent neural network (RNN) applied to embeddings of characters in the token. Such character-level embeddings have been shown to be helpful to deal with out-of-vocab (OOV) tokens. We then use a bi-directional RNN to produce new representation $u_1^Q, \ldots, u_m^Q$ and $u_1^P, \ldots, u_n^P$ of all words in the question and passage respectively:

$$u_t^Q = \text{BiRNN}_Q(u_{t-1}^Q, [e_t^Q, c_t^Q]) \quad (1)$$
$$u_t^P = \text{BiRNN}_P(u_{t-1}^P, [e_t^P, c_t^P]) \quad (2)$$

We choose to use Gated Recurrent Unit (GRU) (Cho et al., 2014) in our experiment since it performs similarly to LSTM (Hochreiter and Schmidhuber, 1997) but is computationally cheaper.

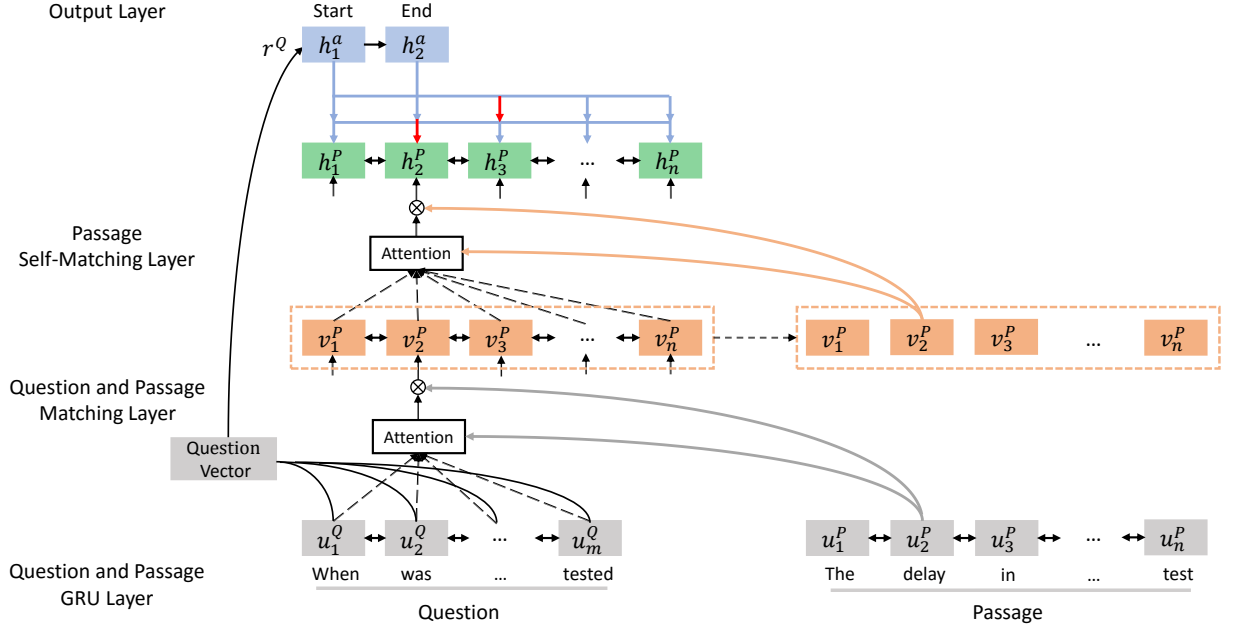

Figure 1: Gated Self-Matching Networks structure overview.

### 3.2 Gated Attention-based Recurrent Networks

We propose a gated attention-based recurrent network to incorporate question information into passage representation. It is a variant of attention-based recurrent networks, with an additional gate to determine the importance of information in the passage regarding a question. Given question and passage representation $\{u_t^Q\}_{t=1}^m$ and $\{u_t^P\}_{t=1}^n$, Rocktäschel et al. (2015) propose generating sentence-pair representation $\{v_t^P\}_{t=1}$ via soft-alignment of words in the question and passage as follows:

$$v_t^P = \text{RNN}(v_{t-1}^P, c_t) \qquad (3)$$

where $c_t = att(u^Q, [u_t^P, v_{t-1}^P])$ is an attention-pooling vector of the whole question ($u^Q$):

$$s_j^t = \text{v}^{\text{T}}\tanh(W_u^Q u_j^Q + W_u^P u_t^P + W_v^P v_{t-1}^P)$$
$$a_i^t = \exp(s_i^t)/\Sigma_{j=1}^m \exp(s_j^t)$$
$$c_t = \Sigma_{i=1}^m a_i^t u_i^Q \qquad (4)$$

Each passage representation $v_t^P$ dynamically incorporates aggregated matching information from the whole question.

Wang and Jiang (2016a) introduce match-LSTM, which takes $u_t^P$ as an additional input into the recurrent network:

$$v_t^P = \text{RNN}(v_{t-1}^P, [u_t^P, c_t]) \qquad (5)$$

To determine the importance of passage parts and attend to the ones relevant to the question, we add another gate to the input ($[u_t^P, c_t]$) of RNN:

$$g_t = \text{sigmoid}(W_g[u_t^P, c_t])$$
$$[u_t^P, c_t]^* = g_t \odot [u_t^P, c_t] \qquad (6)$$

Different from the gates in LSTM or GRU, the additional gate is based on the current passage word and its attention-pooling vector of the question, which focuses on the relation between the question and current passage word. The gate effectively model the phenomenon that only parts of the passage are relevant to the question in reading comprehension and question answering. $[u_t^P, c_t]^*$ is utilized in subsequent calculations instead of $[u_t^P, c_t]$. We call this gated attention-based recurrent networks. It can be applied to variants of RNN, such as GRU and LSTM. We also conduct experiments to show the effectiveness of the additional gate on both GRU and LSTM.

### 3.3 Self-Matching Attention

Through gated attention-based recurrent networks, question-aware passage representation $\{v_t^P\}_{t=1}^n$ is generated to pinpoint important parts in the passage. One problem with such representation is that it has very limited knowledge of context. One answer candidate is often oblivious to important

cues in the passage outside its surrounding window. Moreover, there exists some sort of lexical or syntactic divergence between the question and passage in the majority of SQuAD dataset (Rajpurkar et al., 2016). Passage context is necessary to infer the answer. To address this problem, we propose directly matching the question-aware passage representation against itself. It dynamically collects evidence from the whole passage for words in passage and encodes the evidence relevant to the current passage word and its matching question information into the passage representation $h_t^P$:

$$h_t^P = \text{BiRNN}(h_{t-1}^P, [v_t^P, c_t]) \tag{7}$$

where $c_t = att(v^P, v_t^P)$ is an attention-pooling vector of the whole passage ($v^P$):

$$s_j^t = \text{v}^\text{T}\tanh(W_v^P v_j^P + W_v^{\tilde{P}} v_t^P)$$
$$a_i^t = \exp(s_i^t)/\Sigma_{j=1}^n \exp(s_j^t)$$
$$c_t = \Sigma_{i=1}^n a_i^t v_i^P \tag{8}$$

An additional gate as in gated attention-based recurrent networks is applied to $[v_t^P, c_t]$ to adaptively control the input of RNN.

Self-matching extracts evidence from the whole passage according to the current passage word and question information.

### 3.4 Output Layer

We follow Wang and Jiang (2016b) and use pointer networks (Vinyals et al., 2015) to predict the start and end position of the answer. In addition, we use an attention-pooling over the question representation to generate the initial hidden vector for the pointer network. Given the passage representation $\{h_t^P\}_{t=1}^n$, the attention mechanism is utilized as a pointer to select the start position ($p^1$) and end position ($p^2$) from the passage, which can be formulated as follows:

$$s_j^t = \text{v}^\text{T}\tanh(W_h^P h_j^P + W_\text{h}^a h_{t-1}^a)$$
$$a_i^t = \exp(s_i^t)/\Sigma_{j=1}^n \exp(s_j^t)$$
$$p^t = \arg\max(a_1^t, \dots, a_n^t) \tag{9}$$

Here $h_{t-1}^a$ represents the last hidden state of the answer recurrent network (pointer network). The input of the answer recurrent network is the attention-pooling vector based on current predicted probability $a^t$:

$$c_t = \Sigma_{i=1}^n a_i^t h_i^P$$
$$h_t^a = \text{RNN}(h_{t-1}^a, c_t) \tag{10}$$

When predicting the start position, $h_{t-1}^a$ represents the initial hidden state of the answer recurrent network. We utilize the question vector $r^Q$ as the initial state of the answer recurrent network. $r^Q = att(u^Q, V_r^Q)$ is an attention-pooling vector of the question based on the parameter $V_r^Q$:

$$s_j = \text{v}^\text{T}\tanh(W_u^Q u_j^Q + W_\text{v}^Q V_r^Q)$$
$$a_i = \exp(s_i)/\Sigma_{j=1}^m \exp(s_j)$$
$$r^Q = \Sigma_{i=1}^m a_i u_i^Q \tag{11}$$

To train the network, we minimize the sum of the negative log probabilities of the ground truth start and end position by the predicted distributions.

## 4 Experiment

### 4.1 Implementation Details

We specially focus on the SQuAD dataset to train and evaluate our model, which has garnered a huge attention over the past few months. SQuAD is composed of 100,000+ questions posed by crowd workers on 536 Wikipedia articles. The dataset is randomly partitioned into a training set (80%), a development set (10%), and a test set (10%). The answer to every question is a segment of the corresponding passage.

We use the tokenizer from Stanford CoreNLP (Manning et al., 2014) to preprocess each passage and question. The Gated Recurrent Unit (Cho et al., 2014) variant of LSTM is used throughout our model. For word embedding, we use pretrained case-sensitive GloVe embeddings[2] (Pennington et al., 2014) for both questions and passages, and it is fixed during training; We use zero vectors to represent all out-of-vocab words. We utilize 1 layer of bi-directional GRU to compute character-level embeddings and 3 layers of bi-directional GRU to encode questions and passages, the gated attention-based recurrent network for question and passage matching is also encoded bidirectionally in our experiment. The hidden vector length is set to 75 for all layers. We also apply dropout (Srivastava et al., 2014) between layers with a dropout rate of 0.2. The model is optimized with AdaDelta (Zeiler, 2012) with an initial learning rate of 1.

---

| | Dev Set | Test Set |
|---|---|---|
| *Single model* | EM / F1 | EM / F1 |
| LR Baseline (Rajpurkar et al., 2016) | 40.0 / 51.0 | 40.4 / 51.0 |
| Dynamic Chunk Reader (Yu et al., 2016) | 62.5 / 71.2 | 62.5 / 71.0 |
| Match-LSTM with Ans-Ptr (Wang and Jiang, 2016b) | 64.1 / 73.9 | 64.7 / 73.7 |
| Dynamic Coattention Networks (Xiong et al., 2016) | 65.4 / 75.6 | 66.2 / 75.9 |
| RaSoR (Lee et al., 2016) | 66.4 / 74.9 | - / - |
| BiDAF (Seo et al., 2016) | 68.0 / 77.3 | 68.0 / 77.3 |
| jNet (USTC&National Research Council Canada&York University) | - / - | 68.7 / 77.4 |
| Multi-Perspective Matching (Wang et al., 2016) | - / - | 68.9 / 77.8 |
| **Gated Self-Matching Networks** | **71.1 / 79.5** | **71.3 / 79.7** |
| *Ensemble model* | | |
| Fine-Grained Gating (Yang et al., 2016) | 62.4 / 73.4 | 62.5 / 73.3 |
| Match-LSTM with Ans-Ptr (Wang and Jiang, 2016b) | 67.6 / 76.8 | 67.9 / 77.0 |
| RaSoR (Lee et al., 2016) | 68.2 / 76.7 | - / - |
| Dynamic Coattention Networks (Xiong et al., 2016) | 70.3 / 79.4 | 71.6 / 80.4 |
| BiDAF (Seo et al., 2016) | 73.3 / 81.1 | 73.3 / 81.1 |
| Multi-Perspective Matching (Wang et al., 2016) | - / - | 73.8 / 81.3 |
| **Gated Self-Matching Networks** | **75.6 / 82.8** | **75.9 / 82.9** |
| Human Performance (Rajpurkar et al., 2016) | 80.3 / 90.5 | 77.0 / 86.8 |
| FastQA[*] (German Research Center for Artificial Intelligence) | - / - | 68.4 / 77.1 |
| FastQAExt[*] (German Research Center for Artificial Intelligence) | - / - | 70.8 / 78.9 |

Table 2: The performance of our gated self-matching networks and competing approaches. * indicates that there is no explicit label to suggest whether the model is a single or ensemble model.

| Single Model | EM / F1 |
|---|---|
| **Gated Self-Matching (GRU)** | **71.1 / 79.5** |
| -Character embedding | 70.3 / 78.9 |
| -Gating | 67.9 / 77.1 |
| -Self-Matching | 66.9 / 76.4 |
| -Gating, -Self-Matching | 65.2 / 74.7 |

Table 3: Ablation tests of single model on the SQuAD dev set.

| Single Model | EM / F1 |
|---|---|
| Base model (GRU) | 64.5 / 74.1 |
| **+Gating** | **66.2 / 75.8** |
| Base model (LSTM) | 64.2 / 73.9 |
| **+Gating** | **66.0 / 75.6** |

Table 4: Effectiveness of gated attention-based recurrent networks for both GRU and LSTM.

## 4.2 Main Results

Two metrics are utilized to evaluate model performance: Exact Match (EM) and F1 score. EM measures the percentage of the prediction that matches one of the ground truth answers exactly. F1 measures the overlap between the prediction and ground truth answers which takes the maximum F1 over all of the ground truth answers. The scores on dev set are evaluated by the official script[3]. Since the test set is hidden, we are required to submit the model to Stanford NLP group to obtain the test scores.

Table 2 shows exact match and F1 scores on the dev and test set[4] of our model and competing approaches. As we can see, our method clearly outperforms the baseline and several strong state-of-the-art systems for both single model and ensembles.

## 4.3 Ablation Study

We do ablation tests on the dev set to analyze the contribution of components of gated self-matching networks. As illustrated in Table 3, the gated attention-based recurrent network (GARNN) and self-matching attention mechanism positively contribute to the final results of gated self-matching

---

[3]Downloaded from http://stanford-qa.com

[4]Extracted from SQuAD leaderboard http://stanford-qa.com on Feb. 6, 2017.

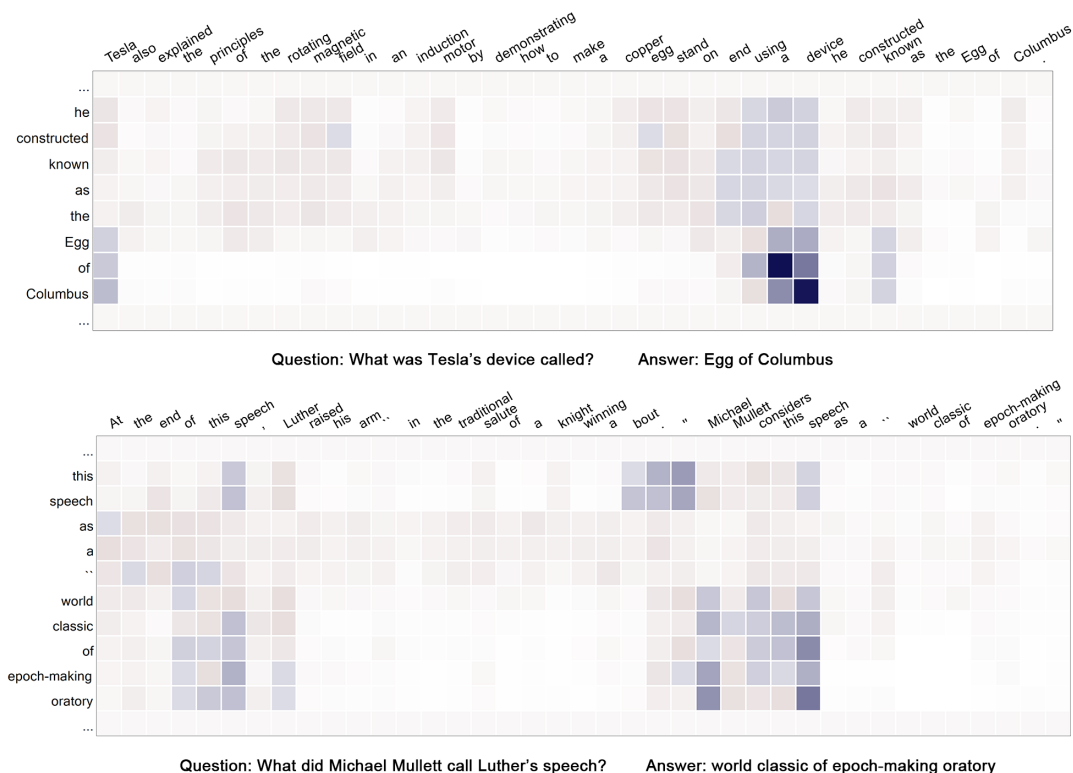

Figure 2: Part of the attention matrices for self-matching. Each row is the attention weights of the whole passage for the current passage word. The darker the color is the higher the weight is. Some key evidence relevant to the question-passage tuple is more encoded into answer candidates.

networks. Removing self-matching results in 4.2 point EM drop, which reveals that information in the passage plays an important role. Character-level embeddings contribute towards the model's performance since it can better handle out-of-vocab or rare words. To show the effectiveness of GARNN for variant RNNs, we conduct experiments on the base model (Wang and Jiang, 2016b) of different variant RNNs. The base model match the question and passage via a variant of attention-based recurrent network (Wang and Jiang, 2016a), and employ pointer networks to predict the answer. Character-level embeddings are not utilized. As shown in Table 4, the gate introduced in question and passage matching layer is helpful for both GRU and LSTM on the SQuAD dataset.

## 5 Discussion

### 5.1 Encoding Evidence from Passage

To show the ability of the model for encoding evidence from the passage, we draw the alignment of the passage against itself in self-matching. The attention weights are shown in Figure 2, in which the darker the color is the higher the

weight is. We can see that key evidence aggregated from the whole passage is more encoded into the answer candidates. For example, the answer "Egg of Columbus" pays more attention to the key information "Tesla", "device" and the lexical variation word "known" that are relevant to the question-passage tuple. The answer "world classic of epoch-making oratory" mainly focuses on the evidence "Michael Mullet", "speech" and lexical variation word "considers". For other words, the attention weights are more evenly distributed between evidence and some irrelevant parts. Self-matching do adaptively aggregate evidence for words in passage.

### 5.2 Result Analysis

To further analyse the model's performance, we show the exact match and F1 score for different question types (Figure 3(a)), different answer lengths (Figure 3(b)), different passage lengths (Figure 3(c)) and different question lengths (Figure 3(d)). As we can see, the type of "what" takes up a majority of questions. Our model is better at "when" and "who" questions, but poorly on "why"

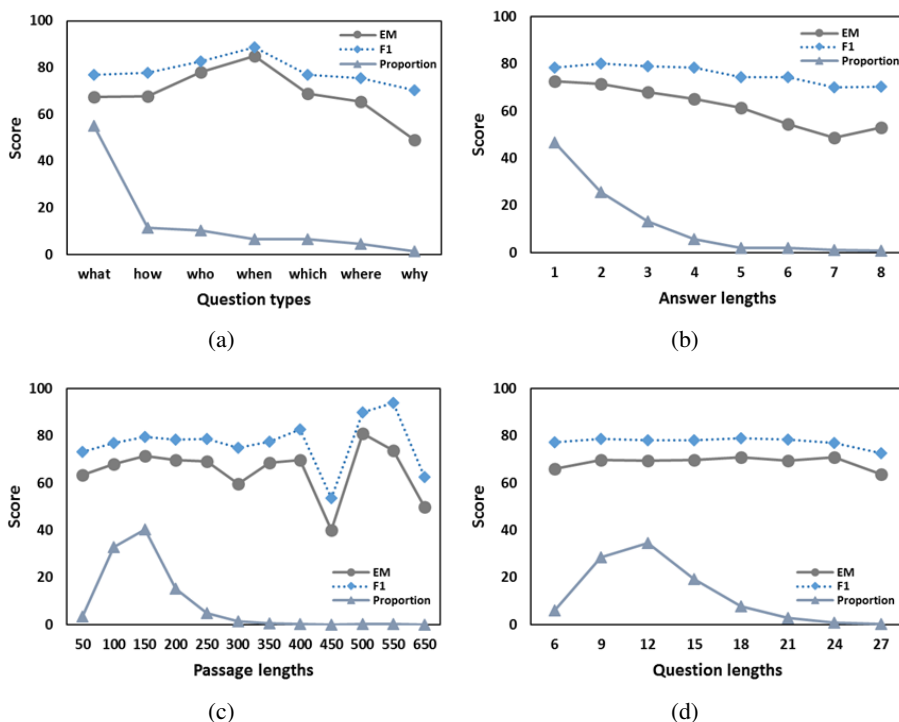

(a)  (b)

(c)  (d)

Figure 3: Model performance on different question types (a), different answer lengths (b), different passage lengths (c), different question lengths (d). The point on the x-axis of figure (c) and (d) represent the datas whose passages length or questions length are between the value of current point and last point.

questions. From the Graph 3(b), the majority of answers are in short length. With the increase of answer length, the performance of our model obviously drops. Besides, the gap between exact match and F1 widens as answer length increases, which indicates our model could locate the core of the answer to some extent. From Graph 3(c) and 3(d), the majority of passages and questions are not too long. Moreover, we discover that the performance remains stable with the increase in length, the obvious drop in longer passage is mainly because the proportion is too small. Our model is largely agnostic to long passages and focuses on important part of the passage.

## 6 Related Work

**Reading Comprehension and Question Answering Dataset** Benchmark datasets play an important role in recent progress in reading comprehension and question answering research. Existing datasets can be classified into two categories according to whether they are manually labeled. Those that are labeled by humans are always in high quality (Richardson et al., 2013; Berant et al., 2014; Yang et al., 2015), but are too small for training modern data-intensive models. Those that

are automatically generated from natural occurring data can be very large (Hill et al., 2016; Hermann et al., 2015), which allow the training of more expressive models. However, they are in cloze style, in which the goal is to predict the missing word (often a named entity) in a passage. Moreover, Chen et al. (2016) have shown that the CNN / Daily News dataset (Hermann et al., 2015) requires less reasoning than previously thought, and conclude that performance is almost saturated.

Different from above datasets, the SQuAD provides a large and high-quality dataset. The answers in SQuAD often include non-entities and can be much longer phrase, which is more challenging than cloze-style datasets. Moreover, Rajpurkar et al. (2016) show that the dataset retains a diverse set of answers and requires different forms of logical reasoning, including multi-sentence reasoning. MS MARCO (Nguyen et al., 2016) is also a large-scale dataset. The questions in the dataset are real anonymized queries issued through Bing or Cortana and the passages are related web pages. For each question in the dataset, several related passages are provided. However, the answers are human generated, which is different from SQuAD where answers must be a span of the passage.

**End-to-end Neural Networks for Reading Comprehension** Along with cloze-style datasets, several powerful deep learning models (Hermann et al., 2015; Hill et al., 2016; Chen et al., 2016; Kadlec et al., 2016; Sordoni et al., 2016; Cui et al., 2016; Trischler et al., 2016; Dhingra et al., 2016; Shen et al., 2016) have been introduced to solve this problem. Hermann et al. (2015) first introduce attention mechanism into reading comprehension. Hill et al. (2016) propose a window-based memory network for CBT dataset. Kadlec et al. (2016) introduce pointer networks with one attention step to predict the blanking out entities. Sordoni et al. (2016) propose an iterative alternating attention mechanism to better model the links between question and passage. Trischler et al. (2016) solve cloze-style question answering task by combining an attentive model with a reranking model. Dhingra et al. (2016) propose iteratively selecting important parts of the passage by a multiplying gating function with the question representation. Cui et al. (2016) propose a two-way attention mechanism to encode the passage and question mutually. Shen et al. (2016) propose iteratively inferring the answer with a dynamic number of reasoning steps and is trained with reinforcement learning.

Neural network-based models demonstrate the effectiveness on the SQuAD dataset. Wang and Jiang (2016b) combine match-LSTM and pointer networks to produce the boundary of the answer. Xiong et al. (2016) and Seo et al. (2016) employ variant coattention mechanism to match the question and passage mutually. Xiong et al. (2016) propose a dynamic pointer network to iteratively infer the answer. Yu et al. (2016) and Lee et al. (2016) solve SQuAD by ranking continuous text spans within passage. Yang et al. (2016) present a fine-grained gating mechanism to dynamically combine word-level and character-level representation and model the interaction between questions and passages. Wang et al. (2016) propose matching the context of passage with the question from multiple perspectives.

Different from the above models, we introduce self-matching attention in our model. It dynamically refines the passage representation by looking over the whole passage and aggregating evidence relevant to the current passage word and question, allowing our model make full use of passage information. Weightedly attending to word context

has been proposed in several works. Ling et al. (2015) propose considering window-based contextual words differently depending on the word and its relative position. Cheng et al. (2016) propose a novel LSTM network to encode words in a sentence which considers the relation between the current token being processed and its past tokens in the memory. Parikh et al. (2016) apply this method to encode words in a sentence according to word form and its distance. Since passage information relevant to question is more helpful to infer the answer in reading comprehension, we apply self-matching based on question-aware representation and gated attention-based recurrent networks. It helps our model mainly focus on question-relevant evidence in the passage and dynamically look over the whole passage to aggregate evidence.

Another key component of our model is the attention-based recurrent network, which has demonstrated success in a wide range of tasks. Bahdanau et al. (2014) first propose attention-based recurrent networks to infer word-level alignment when generating the target word. Hermann et al. (2015) introduce word-level attention into reading comprehension to model the interaction between questions and passages. Rocktäschel et al. (2015) and Wang and Jiang (2016a) propose determining entailment via word-by-word matching. The gated attention-based recurrent network is a variant of attention-based recurrent network with an additional gate to model the fact that passage parts are of different importance to the particular question for reading comprehension and question answering.

## 7 Conclusion

In this paper, we present gated self-matching networks for reading comprehension and question answering. We introduce the gated attention-based recurrent networks and self-matching attention mechanism to obtain representation for the question and passage, and then use the pointer-networks to locate answer boundaries. Our model achieves state-of-the-art results on the SQuAD dataset, outperforming several strong competing systems. As for future work, we are applying the gated self-matching networks to other reading comprehension and question answering datasets, such as the MS MARCO dataset (Nguyen et al., 2016).

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
