# Peer review of "Gated Self-Matching Networks for Reading Comprehension and Question Answering"

_ACL 2017 — decision unknown_

[Official Review · Reviewer 1 · rating 4 · confidence 2]
soundness 5 · originality 5 · clarity 4 · impact 3 · substance 4 · appropriateness 5 · meaningful comparison 3 · presentation format Oral Presentation

This work describes a gated attention-based recurrent neural network method for
reading comprehension and question answering. This method employs a
self-matching attention technique to counterbalance the limited context
knowledge of gated attention-based recurrent neural networks when processing
passages. Finally, authors use pointer networks  with signals from the question
attention-based vector to predict the beginning and ending of the answer.
Experimental results with the SQuAD dataset offer state-of-the-art performance
compared with several recent approaches. 

The paper is well-written, structured and explained. As far as I know, the
mathematics look also good. In my opinion, this is a very interesting work
which may be useful for the question answering community.

I was wondering if the authors have plans to release the code of this approach.
From that perspective, I miss a bit of information about the technology used
for the implementation (theano, CUDA, CuDNN...), which may be useful for
readers.

I would appreciate if authors could perform a test of statistical significance
of the results. That would highlight even more the quality of your results.

Finally, I know that the space may be a constraint, but an evaluation including
some additional dataset would validate more your work.

[Official Review · Reviewer 2 · rating 4 · confidence 4]
soundness 5 · originality 5 · clarity 4 · impact 3 · substance 4 · appropriateness 5 · meaningful comparison 3 · presentation format Oral Presentation

This paper presents the gated self-matching network for reading comprehension
style question answering. There are three key components in the solution: 

(a) The paper introduces the gated attention-based recurrent network to obtain
the question-aware representation for the passage. Here, the paper adds an
additional gate to attention-based recurrent networks to determine the
importance of passage parts and attend to the ones relevant to the question.
Here they use word as well as character embeddings to handle OOV words.
Overall, this component is inspired from Wang and Jiang 2016.

(b) Then the paper proposes a self-matching attention mechanism to improve the
representation for the question and passage by looking at wider passage context
necessary to infer the answer. This component is completely novel in the paper.

(c) At the output layer, the paper uses pointer networks to locate answer
boundaries. This is also inspired from Wang and Jiang 2016

Overall, I like the paper and think that it makes a nice contribution.

- Strengths:

The paper clearly breaks the network into three component for descriptive
purposes, relates each of them to prior work and mentions its novelties with
respect to them. It does a sound empirical analysis by describing the impact of
each component by doing an ablation study. This is appreciated.

The results are impressive!

- Weaknesses:

The paper describes the results on a single model and an ensemble model. I
could not find any details of the ensemble and how was it created. I believe it
might be the ensemble of the character based and word based model. Can the
authors please describe this in the rebuttal and the paper.

- General Discussion:

Along with the ablation study, it would be nice if we can have a
qualitative analysis describing some example cases where the components of
gating, character embedding, self embedding, etc. become crucial ... where a
simple model doesn't get the question right but adding one or more of these
components helps. This can go in some form of appendix or supplementary.